# Effects of Estrogens on Platelets and Megakaryocytes

**DOI:** 10.3390/ijms20123111

**Published:** 2019-06-25

**Authors:** Marion Dupuis, Sonia Severin, Emmanuelle Noirrit-Esclassan, Jean-François Arnal, Bernard Payrastre, Marie-Cécile Valéra

**Affiliations:** 1Inserm, U1048 and Université Toulouse III, I2MC, Toulouse 31432, France; dupuis.marion@free.fr (M.D.); sonia.severin@inserm.fr (S.S.); noirrit.e@odonto-tlse.eu (E.N.-E.); jean-francois.arnal@inserm.fr (J.-F.A.); bernard.payrastre@inserm.fr (B.P.); 2CHU de Toulouse, Laboratoire d’Hématologie, Toulouse 31059, France

**Keywords:** estrogens, platelets, megakaryocytes

## Abstract

In women, oral menopausal hormonal therapy (MHT) is associated with adverse effects including an increased incidence of thromboembolic events, classically attributed to an increase in several liver-derived coagulation factors due to hepatic first pass. While platelets are central players in thrombus constitution, their implication in women treated with estrogens remains incompletely characterized. Platelets and their medullar progenitors, megakaryocytes, express estrogen receptors (ER) that may explain, at least in part, a sensitivity to hormonal changes. The purpose of this review is to summarize our current knowledge of estrogen actions on platelets and megakaryocytes in mice following in vivo administration and in women using MHT.

## 1. Introduction

Menopause is defined as a permanent cessation of menstruation and ovulation due to ovarian aging. In women, the decline in progesterone and estrogen levels can lead to a number of bothersome symptoms such as vasomotor hot flushes, night sweats, insomnia, and genitourinary disturbances. Menopause is also associated with an increased risk of cardiovascular events and metabolic disorders and an elevated risk of developing osteoporosis and fractures. These symptoms could be alleviated by the administration of menopausal hormonal therapy (MHT), at first based on oral administration of natural estrogens extracted from the urine of pregnant mares (mainly in the United States), and later from the synthesis of the natural estrogen, 17β-estradiol (E2), especially using the transdermal route (mainly in Europe). The addition of a progestogen to estrogens therapy among postmenopausal women with an intact uterus is necessary to prevent the increased risk of estrogen-induced endometrial hyperplasia and adenocarcinoma [1]. The Women’s Health Initiative (WHI) study investigated the potential risks and benefits associated with oral MHT by including around 16,000 postmenopausal women treated with conjugated estrogens (CE) in combination with medroxyprogesterone acetate (MPA). This hormonal combination was associated with an increased incidence of thrombotic events, coronary disease, and breast cancer [2]. In hysterectomized women given either CE alone or a placebo, a tendency towards an increased risk of venous thromboembolism (VTE) and a significantly higher risk of stroke were also observed [3]. The addition of progestogens was associated with additional increases in the risk of VTE according to the type of molecules [4]. In addition, several observational studies suggest that, contrary to the transdermal route, the oral administration of estrogen is a major determinant of the increase in thrombotic events with a higher risk of VTE under oral MHT [5,6,7]. This increased risk appears to be attributable to the impact of the route of estrogen administration leading to an hepatic first pass, and therefore to an increase in some circulating blood coagulation factors [8]. Even if platelets are key players in hemostasis and thrombosis, the impact of hormonal treatments on platelet reactivity is much less described.

Estrogens, and more particularly E2, have pleiotropic effects due to a large tissue distribution of their receptors. The biological effects of estrogens are mediated by their binding to the two major estrogen receptors (ER) [9,10,11]: ERα and ERβ, leading to conformational changes, dimerization, and recruitment of coactivators into the nucleus where they link with estrogen response elements or other transcription factors to regulate the transcription of target genes. Ligand-induced transcriptional activity of ER involves the action of two distinct activation functions (AF), AF1 and AF2. Between estrogen administration and apparent transcriptional results, the time lag is typically in the order of hours to days. However, in addition to these nuclear, also called genomic actions of ER, estrogens have been found to induce rapid effects occurring within minutes following administration. These effects are mediated through receptors associated with the plasma membrane, a process termed “nongenomic”, “membrane-initiated steroid signaling” (MISS), or “extranuclear” effects [12]. ERs have different distributions in tissues with different effects. Platelets and their medullar precursors, megakaryocytes (MK), express ERs [13,14,15,16,17]. Estrogens can also exert rapid effects through a member of the 7-transmembrane G protein-coupled receptor family, G protein-coupled estrogen receptor (GPER, formerly known as GPR30). GPER was not detected in mature MK but was expressed in human hematopoietic stem cells [18].

Progestogens exert their physiological and biological effects by interacting with intracellular progesterone receptors A (PR-A) and B (PR-B). PRs belong to the nuclear receptor superfamily of transcription factors and regulate gene expression following hormone binding. A “non-classical” mechanism mediated by membrane-associated receptors has been described and the progesterone receptor membrane component 1 was detected in human platelet lysates [19].

In this review, we will summarize our current knowledge of estrogen actions on platelets and MK in mice and women following in vivo administration.

## 2. Effects of Estrogens Treatment on Megakaryopoiesis and Platelet Production

Megakaryopoiesis is a complex and specialized process of hematopoietic cell maturation occurring in the bone marrow under the influence of thrombopoietin. Hematopoietic progenitors undergo nuclear (endomitosis with cytokinesis) and cytoplasmic (acquisition of organelles and development of demarcation membrane system) maturation, which ends with proplatelet extension and platelet release in blood circulation.

### 2.1. Effect of Chronic Estrogens Treatment in Mice

A sex difference has been evidenced in the number of MK in the splenic red pulp but not in the bone marrow [20]. In female mice, physiological E2 level had no effect on MK count in the bone marrow [21]. A supraphysiological subcutaneous E2 dose (10 or 100 μg/kg) has been shown to induce a decrease in MK number by day 10 compared to vehicle-treated ovariectomized mice. However, a single higher injection of E2 (500 μg/kg) has been shown to induce an initial increase in mature MK number in the bone marrow 2 days after the injection, followed by a gradual reduction in the number of MK reaching 75% decrease 12 days after injection [21,22]. In addition, more mature MK (with increased size and polylobulated nuclei) were found in the bone marrow of mice treated with estrogen via intraperitoneal injection (500 μg/kg/day) for three days, accompanied by a significant increase in the platelet count in the blood [23]. The impact of these high doses of E2 on megakaryopoiesis reported in these experiments could be attributed to (i) an effect of E2 on the bone, (ii) a greater differentiation of precursors into MKs, or (iii) a suppression of apoptosis of mature MK [21].

Fox et al. have shown a decrease in the platelet count in ovariectomized mice receiving E2 (10 or 100 μg/kg/day by subcutaneous injections) for 10 days [21]. Our team has reported that a three-week E2 treatment of ovariectomized mice (200 μg/kg/day) slightly reduced the platelet count [24]. In mice treated by a chronic subcutaneous administration of E2 (80 μg/kg/day) alone or in addition to progesterone (P4, pellet of 10 mg) during the three weeks, platelet count was not significantly affected in comparison to ovariectomized mice [25]. In addition, thrombocytopenia was induced after a three-week treatment by intraperitoneal injection of antimouse GPIbαantibody. The platelet count recovery was measured fifteen days after this immune-induced thrombocytopenia and was similar in ovariectomized mice, E2, and E2 + P4 treated-mice. In conclusion, platelet production did not appear to be impacted by the treatments [25].

At variance, a slightly higher dose of E2 (500 μg/kg) increased the platelet number two days after a single subcutaneous injection [21]. The data from the literature concerning the impact of estrogen treatment on MK and platelet production in mice are compiled in Table 1. Duration of treatment and the total dose received impact the number of cells. In this context, the day of observation seems to be crucial: After a chronic administration, the results at the final time must be preferred. 

### 2.2. Effect of Estrogens Therapy in Women

The literature on the role of E2 on megakaryopoiesis and platelet production in women remains scarce. Bord et al. have shown a higher number of medullar MK after a long term oral or transdermal conventional MHT without modification of the total bone marrow cell number [26]. An increased medullar MK number was also observed after a long-term high dose of E2 in women who had undergone total hysterectomy and salpingectomy (at least 14 months before) in comparison to premenopausal women [26]. In the same study, the authors have shown that the administration of E2 increased the synthesis of TGFβ by MK [26]. As TGFβ is a growth factor implicated in osteoblast differentiation, this result suggests that E2 and MK may modulate bone remodeling.

A study of 38 postmenopausal women treated with oral MHT has shown a slight increase in the platelet count and platelet volume six weeks after the beginning of the treatment [27]. The Kronos Early Estrogen Prevention Study (KEEPS) has compared the effects of oral CE and transdermal E2 in menopausal women [28]. In this study, the platelet count did not differ significantly among the groups at baseline and after 48 months of treatment. A study including a small number of women also found no modification of the platelet count after three months of an oral, continuously or transdermal cyclical and sequential, E2 treatment [29]. However, a significant decrease in the platelet count three months after an E2 transdermal treatment was reported in another study [30]. The data from the literature concerning the impact of estrogen treatment on MK and platelet production in women are compiled in Table 1.

To summarize these data, one could say that the effects of hormonal therapy on the platelet count appear to be limited, despite some differences according to the type of estrogen and progestogen used, the route of administration, the duration of the treatment, and the inclusion criteria of the women included in the studies. The mechanisms underlying the effect of estrogen on megakaryopoiesis remain unknown. The study of Du et al. suggests that the action of estrogens on human cord blood-derived CD34^+^ cells may be mediated by ERβ-dependent transactivation of GATA1 and subsequent activation of STAT1 [23]. To our knowledge, the effect of estrogen on platelet life span is unknown.

## 3. Effects of Estrogen Treatment on Platelet Activation

Following a vascular injury, platelets first transiently adhere to collagen-bound von Willebrand factor (vWF) via its GP1b-V-IX receptor and then firmly adhere to the collagen matrix via its GPVI receptor and the α2β1 integrin [31,32]. Following this activation pathway, platelets secrete important molecules including ADP, serotonin, vWF, and fibrinogen, and synthesize thromboxane A2 (TXA2). This allows the recruitment of circulating platelets to form the thrombus via platelet aggregation through binding of fibrinogen to the activated αIIbβ3 integrin. Conflicting results have been reported concerning the effects of administration of estrogen on platelet activation either in mice or in women.

### 3.1. Effect of Estrogen Treatment on Platelet Aggregation Response in Mice

A study published in 1985 has shown that mice treated by an estrogen pellet containing 0.5 mg of E2 and then submitted to a mesenteric endothelial injury exhibited a faster thrombosis response compared to the placebo group [33]. Similar results were found after a cerebral vessels injury induced by a noxious light/dye stimulus [34]. In contrast, we recently reported that a three-week high dose of E2 (200 μg/kg/day) treatment increased the tail-bleeding time, and protected animals against collagen/epinephrine-induced thromboembolism, and against injury-induced carotid artery thrombosis compared to ovariectomized or sham-operated mice [24]. In addition, washed platelets, isolated from these E2-treated mice, displayed a reduced aggregation response following stimulation by thrombin, collagen, or thromboxane A2 analogue [24]. Ex vivo thrombus formation assays conducted in whole blood under physiological flow conditions on a collagen matrix confirmed that platelets from chronic E2-treated mice were less efficient at forming thrombi compared to control platelets [24]. Platelet granule secretion assessed by P-selectin expression measurement and platelet shape change were, however, not significantly affected by E2 treatment. Using 2D-DIGE coupled to mass spectrometry analysis, it was shown that E2 treatment decreased the expression of several platelet proteins including β1-tubulin, a major constituent of microtubules known to modulate platelet production and function [24]. Moreover, Geng et al. have shown that the decrease of aggregation response was associated with a reduction of platelet and MK GPVI expression after E2 therapy in ovariectomized mice [35].

Mouse models targeting ERs were developed to evaluate the relative contribution of each receptor, i.e., ERα (ERαKO mice) and ERβ (ERβKO mice) or activation functions, i.e., AF1 (ERα–AF1^0^) [12]. Hematopoietic chimera mice harboring a selective deletion of ERs α or β were used to demonstrate that the effects of a chronic administration of E2 (200 μg/kg/day, three weeks) on tail- bleeding time and thromboembolism were exclusively because of hematopoietic ERα [24]. The activation function 1 domain was not required for resistance to thromboembolism. A new knock-in mouse model has been recently generated by mutating the cysteine 451 palmitoylation site of ERα to alanine (designated C451A-ERα). This model provided a specific loss of function of membrane ERα and was used to assess the particular role of membrane versus nuclear action of ERα in vivo [36]. Using combined genetic and pharmacological approaches, we observed that the prolongation of the tail-bleeding time after a chronic administration of E2 was not prevented by the absence of either membrane or nuclear ERα activation in bone marrow, suggesting a redundancy of these two functions for this effect. In addition, hematopoietic membrane ERα was neither sufficient nor necessary to protect E2-treated mice from collagen/epinephrine-induced thromboembolism but the protection was significantly reduced in the absence of hematopoietic nuclear ERα activation [37]. These studies demonstrate that hematopoietic cells (such as megakaryocytes and immune cells) constitute a crucial target in the antithrombotic effects of estrogens.

A three-week chronic administration of E2 (80 μg/kg/day) combined with P4 (pellet of 10 mg) led to increased tail-bleeding time, resistance to collagen/epinephrine-induced thromboembolism, and protection against thrombosis in two different models: Occlusive thrombus formation in the right carotid artery following FeCl3-induced injury, and venous thrombosis induced by inferior vena cava stasis after ligation [25]. E2 combined with P4 protected mice from thrombosis independently of functional deficiencies in coagulation. In E2- and E2 + P4-treated mice, the tail bleeding time was increased [25]. Freudenberger et al. demonstrated that, in ApoE-deficient mice given a Western diet, a long-term treatment (90 days) with medroxyprogesterone acetate (MPA) alone or MPA + E2 increased arterial thrombosis following photochemical injury of the right carotid artery [38], whereas another synthetic progestin, norethisterone acetate, did not impact arterial thrombosis [39]. In mice, the type of progestogen added is of importance in terms of impact on thrombosis.

Estetrol (E4) is a natural estrogen synthesized exclusively during pregnancy by the human fetal liver and reaches the maternal circulation through the placenta. A chronic treatment of E4 (6 mg/kg/d) increased the tail-bleeding time of ovariectomized mice and protected them from both arterial and venous thrombosis. First, a protection of E4-treated mice against thrombus formation induced by FeCl3 injury of the carotid artery was observed. Second, using a well-established ex vivo flow-based thrombus formation assay, E4 treatment significantly decreased platelet thrombus growth on collagen at arterial shear rate. E4 had no effect on the stability of the thrombus, even at high pathological shear rate, but rather reduced the capacity of platelets to form a growing thrombus under flow. In contrast, significant aggregation defects of washed platelet following stimulation in suspension was not detected [40].

Altogether, as highlighted in Table 2, these studies strongly suggest that estrogens can modulate platelet activation in mice although the molecular mechanisms are still incompletely characterized. The results are dependent upon the duration of treatment, the total dose received by mice, and the route of administration of estrogens (and progestogens). In addition, contradictory results can also be explained by the use of different models of thrombosis and mice strains.

### 3.2. Effect of Estrogens Therapy on Platelet Activation in Women

Several studies have shown a gender difference in the properties of platelets, suggesting a role played by sex hormones [41,42,43,44]. In women, many characteristics of platelets have been shown to vary in function depending on the phase of the menstrual cycle [44,45,46]. Tarantino et al. have demonstrated a periodic fluctuation in platelet adhesion to collagen with a biphasic aspect correlated to the peak levels of plasmatic E2 [15]. Moreover, it has been shown that platelets have a higher affinity to fibrinogen during the luteal phase of the cycle compared to the follicular phase [42]. In addition, an impact of menopause on platelet activation has been found but the results remain controversial. The levels of activated GPIIb-IIIa and of P-selectin in postmenopausal women was higher than in premenopausal women [47,48] but Aldrighi et al. suggested a lower platelet activation status after menopause [49]. In another study, no difference was observed [50].

In 15 postmenopausal women, levels of activated αIIbβ3 integrin and of P-selectin expression reflecting platelet activation were significantly reduced after six months of MHT [48]. Conversely, in a randomized, placebo-controlled study, twelve weeks of combined oral MHT was associated with an increase in platelet activation parameters (P-selectin and glycoprotein 53), suggesting alpha granule and lysosome degranulation. In women treated with E2 only, an increase in P-selectin expression was observed [51]. The increase of P-selectin expression in stimulated platelets was more important when E2 treatment was transdermal and the addition of progestin had no impact [52]. However, the activation of platelets, measured by thromboxane B2 level in whole blood, was reduced after twelve months of oral or transdermal MHT [53]. Rank et al. have shown that six months after a MHT treatment, there was a higher plasma level of micro particles derived from activated platelets, compared to the control group [54].

Bar et al. have found that three months after starting MHT (Premarin^®^, 0.625 mg/day with medroxyprogesterone acetate, 5 mg/day for 14 days per month), there was a significant decrease in adrenaline-induced platelet aggregation and adenosine triphosphate (ATP) release compared to untreated women [55]. Other studies have found no effect of MHT on platelet aggregation stimulated by different agonists [29,56,57]. Women enrolled in the KEEPS study, treated with E2, did not exhibit modifications in platelet aggregation 48 months after starting the treatment, although a tendency towards an increase of platelet secretion was observed [28,58]. In the KEEPS study, it was noted that the platelet content of serotonin was higher in women with oral or transdermal treatment compared to the placebo group [58].

More recently, a multiple-rising-dose, partly randomized, and open-label study has been conducted in postmenopausal women to assess the tolerance, safety, and pharmacokinetics of E4, including its effect on endometrium, vaginal cytology, and the number of hot flashes or sweating. The results showed that E4 had estrogenic effects on hot flashes and on reproductive tissues, indicating its relevance for the treatment of vulvo-vaginal atrophy [59,60]. Interestingly, E4 treatment did not appear to increase the level of hepatic-derived coagulation factors, and therefore, might not increase the risk of thromboembolic events [59].

As highlighted in Table 3, several studies have compared platelet reactivity in postmenopausal women, with or without MHT, but the results are largely contradictory, probably because of the use of different molecules, dosage, time, and route of administration, and the criteria of inclusion of women (age, hormonal treatment). The molecular mechanisms underlying the effects of MHT on platelet activation have been poorly investigated. Some studies have, however, investigated platelet reactivity after an in vitro administration of E2. For instance, the study of Bharat et al. has shown that incubation of platelets with E2 does not modify the baseline of intracellular [Ca^2+^] but reduces ADP-induced [Ca^2+^] response [61]. Platelet ATP release has been shown to be strongly reduced after incubation with E2 (10^−8^ mol/L) [62]. Moro et al. have reported that administration of E2 on platelets in vitro caused a rapid phosphorylation of the tyrosine kinases Src and Pyk2 and the formation of a signaling complex including Src, Pyk2, and phosphoinositide 3-kinase [63].

## 4. Conclusions

Oral MHT has been associated with an increased risk of thromboembolic events [64]. The prothrombotic effects of estrogen are classically attributed to changes in hepatic-derived coagulation factors. Platelets have a central role not only in arterial diseases, but also in venous thrombosis. Changes in platelet number or reactivity in response to physiological levels of estradiol do not appear to influence thrombosis in mouse models. In contrast, chronic high levels of estrogens, reminiscent to those found during gestation, are able to prevent experimental thrombosis in mice. Whether these mechanisms of protection also occur in pregnant women remains to be determined, although our study in a small cohort suggests a reduced capacity of platelets to form a thrombus under high shear stress [65].

Overall, this review illustrates the difficulties of currently making a clear picture of the impact of estrogens on platelet production and functions in vivo and show that further studies are needed to characterize the potential link between increased risks of thrombotic events during MHT and the modulation of platelet responsiveness.

## Figures and Tables

**Table 1 ijms-20-03111-t001:** Summary of the impact of an estrogens treatment on megakaryopoiesis and platelet production. The arrows indicate the changes in megakaryocytes (MK) number and platelet count.

	MK Number	Platelet Count
Mice	Fox et al.: 10 or 100 µg/kg: ↘	Fox et al.: 10 or 100 μg/kg/day (10 days): ↘
	500 μg/kg: ↗	500 μg/kg (single injection): ↗
	Perry et al.: 500 μg/kg: ↗ and ↘	Valera et al. (2017): 80 μg/kg/day (3 weeks): =
	Du et al.: 500 μg/kg/day ↗	Valera et al. (2012): 200 μg/kg/day (3 weeks): ↘
		Du et al.: 500 μg/kg/day (3 days): ↗
Women	Bord et al.: ↗	Ranganath et al. (6 weeks): ↗
		Miller et al. (48 months): =
		Kaplan et al. (3 months): =
		Stachowiak et al. (3 months): ↘

**Table 2 ijms-20-03111-t002:** Impact of subcutaneous estrogens treatment on platelet activation in mice. The arrows indicate the changes in platelet responses. 17β estradiol: E2. Progesterone: P4; medroxyprogesterone acetate: MPA.

	Molecule	Dose and Duration	Results
Rosenblum et al. [33]	E2	Pellet 0.5 mg, 12 days	Faster thrombosis (mesenteric arterioles)
Rosenblum et al. [34]	E2	Pellet 0.5 mg, 12 days	Faster thrombosis (cerebral microvessels)
Valera el al. [20]	E2	Pellet 200 μg/kg/day, 3 weeks	Protection against thromboembolism
			Aggregation and adhesion under flow ↘
Valera et al. [24]	E2 + P4	Pellet E2 80 μg/kg/day,	Protection against venous and arterial thrombosis
		P4 10 mg, 3 weeks	
Freudenberger et al. [38]	E2 + MPA	MPA 27.7 µg/d E2: 1.1µg/d, 90 days	arterial thrombosis ↗
Geng et al. [35]	E2	1-μg 21-day slow-release pellet, 21 days	Aggregation ↘
Valera el al. [25]	Estetrol	Pellet 6 mg/kg/day, 3 weeks	Protection against thromboembolism
			Adhesion under flow ↘

**Table 3 ijms-20-03111-t003:** Impact of menopausal hormonal therapy on platelet activation in women.

Decreased platelet activation	Gu et al.
	Aune et al.
	Bar et al.
No change	Kaplan et al.
	Teede et al.
	Williams et al.
	Miller et al.
Increased platelet activation	Thijs et al.
	Garcia-Martinez et al.
	Miller et al.
	Raz et al.
	Rank et al.

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
