# Peer review of "Effects of Estrogens on Platelets and Megakaryocytes"

_ijms, 2019, doi:10.3390/ijms20123111_

Round 1

Reviewer 1 Report

In this manuscript, Dupuis and colleagues describe as, during MHT, variations of estrogen concentration can affect the properties of platelets and megakaryocytes in postmenopausal women. They highlight how there are many conflicting data in literature probably due to the different molecules used in MHT.

This manuscript is well written, fluent and clear. I have only some little revisions to ask.

1) Into the introduction the authors describe that the WHI investigated the potential risks and benefits associated with oral MHT (CE and MPA). This hormonal combination was associated with an increased incidence of thrombotic events, coronary disease and breast cancer . There are observations about the effects of a MHT with the only estradiol? Is the frequency of thrombotic events the same between women that use menopausal estrogen therapy and those treated with estrogen and progesterone menopausal therapy? Is the progesterone receptor  alsoexpressed in platelets and megacaryocytes?

Lanes 52-55: the authors sustain that a "subpopulation" of steroid receptors mediates are responsible of "non genomic" actions. This is incomplete because also the classical steroid receptors are responsible for the non genomic action of steroid receptors as indicated in 

J Cell Commun Signal. 2010 Dec;4(4):161-72. doi: 10.1007/s12079-010-0103-1. 

Lane 71: is 2 days the classical maturation time for MKs? Please specify to permit a rapid comprehension of the effect induced by estrogens.

I understand that is too hard discuss about the conflicting data in literature but I think that the authors should discuss in more depth, adding their opinion and looking for some common thread. 

Author Response

The specific requirements of the two reviewers were addressed as follows (Minor changes made in the text are in yellow):

REVIEWER 1:

1) Into the introduction the authors describe that the WHI investigated the potential risks and benefits associated with oral MHT (CE and MPA). This hormonal combination was associated with an increased incidence of thrombotic events, coronary disease and breast cancer . There are observations about the effects of a MHT with the only estradiol?

Recent data about the impact of CE and estradiol are summarized in Pickar et al, 2017 :  the safety of HT with regard to cardiovascular disease, thrombosis, the endometrium, the breast, and cognition was reviewed. Differential safety effects of estradiol versus conjugated equine estrogens, and progesterone versus synthetic progestins were reported.Pickar JH, Archer DF, Kagan R, Pinkerton JV, Taylor HS. Safety and benefit considerations for menopausal hormone therapy. Expert Opin Drug Saf. 2017 Aug;16(8):941-954.

Especially about the risk of thrombosis, Laliberté et al, (2018) observed that the incidence of venous thromboembolism remained significantly lower for estradiol transdermal system users than for oral estrogen-only hormone therapy users. This new reference is added (7 ) line 41.

7 : Laliberté F, Dea K, Duh MS, Kahler KH, Rolli M, Lefebvre P.Does the route of administration for estrogen hormone therapy impact the risk of venous thromboembolism? Estradiol transdermal system versus oral estrogen-only hormone therapy. Menopause. 2018 Nov;25(11):1297-1305.

Is the frequency of thrombotic events the same between women that use menopausal estrogen therapy and those treated with estrogen and progesterone menopausal therapy?

The effects of the addition of progestogens on thromboembolic disorders were detailed in an article based on observational studies (Canonico et al, 2011): there is an increased risk of VTE in women using estrogens and progestogens as compared with users of oral estrogens alone. In addition, the type of progestogen has recently emerged as an important determinant of the VTE risk among HRT users. The manuscript has been edited and this reference has been added (4), line 37-38

4 : Canonico M, Plu-Bureau G, Scarabin P-Y. Progestogens and venous thromboembolism among postmenopausal women using hormone therapy. Maturitas. déc 2011;70(4):354‑60.

Is the progesterone receptor also expressed in platelets and megacaryocytes?

This interesting point is now discussed as follow lines 61 to 65:

“Progestogens exert their physiological and biological effects by interacting with intracellular progesterone receptors A (PR-A) and B (PR-B). PRs belong to the nuclear receptor superfamily of transcription factors and regulate gene expression following hormone binding. A “non‐classical’ mechanisms mediated by membrane‐associated receptors has been described and the progesterone receptor membrane component 1 was detected in human platelet lysates (19).”

A new reference has been added (19) :

19 : Fan X, Chen X, Wang C, Dai J, Lu Y, Wang K, et al. Drospirenone enhances GPIb-IX-V-mediated platelet activation. J Thromb Haemost. oct 2015;13(10):1918‑24.

In addition, according to Khetawat et al., no PR messenger RNA or protein was detected in the megakaryocyte lineage but megakaryocytes were generated ex vivo from normal human CD34(+) stem cells.

Khetawat G, Faraday N, Nealen ML, Vijayan KV, Bolton E, Noga SJ, Bray PF. Human megakaryocytes and platelets contain the estrogen receptor beta and androgen receptor (AR): testosterone regulates AR expression. Blood. 2000 Apr 1;95(7):2289-96.

Lanes 52-55: the authors sustain that a "subpopulation" of steroid receptors mediates are responsible of "non genomic" actions. This is incomplete because also the classical steroid receptors are responsible for the non genomic action of steroid receptors as indicated in

J Cell Commun Signal. 2010 Dec;4(4):161-72. doi: 10.1007/s12079-010-0103-1.

The term “subpopulation” has been removed (line 54)

Lane 71: is 2 days the classical maturation time for MKs? Please specify to permit a rapid comprehension of the effect induced by estrogens.

The authors observed MK in bone marrow (femora) 2 days after an injection of E2. The effects of an administration of E2 can be rapid (lines 49-60) in the order of minutes to days but we agree with the reviewer that in the context of a chronic treatment, final times should be privileged.  The manuscript has been edited line 101 to 103.

I understand that is too hard discuss about the conflicting data in literature but I think that the authors should discuss in more depth, adding their opinion and looking for some common thread

The manuscript has been edited. Please read lines 101, 123 and 199.

Reviewer 2 Report

The authors present a paper addressing the effects of estrogens on platelets and megakaryocytes. The theme is interesting and timely but the paper has some flaws that need to be addressed before it merits publication. Specifically:

The authors describe the effects of estrogen through their receptors, but they only mention ER alpha and beta and not GPR30 which is widely studied in hemopoiesis. This papers should be revised including the role of this receptor in platelets and megakaryocytes.

Also, the authors mention that "Hematopoietic chimera mice harboring a selective deletion of ERs alpha or beta were used to demonstrate that the effects of a chronic administration of E2 (200μg/kg/day, 3 weeks) were exclusively because of hematopoietic ERalpha". This sentence is confusing since the major estrogen receptor in the hematopoietic system is ER beta. There are papers that refer ER beta as the key player in megakaryocytes (e.g.  Leukemia volume31pages945956 (2017)). The authors should discuss their results at the light of more recent literature including this paper.

Minor points:

Line 61: "ovarian failure" should be "ovarian aging" or "...due to natural depletion and aging of the finite amount of oocytes".

Author Response

 REVIEWER 2:

The authors present a paper addressing the effects of estrogens on platelets and megakaryocytes. The theme is interesting and timely but the paper has some flaws that need to be addressed before it merits publication.

Specifically: The authors describe the effects of estrogen through their receptors, but they only mention ER alpha and beta and not GPR30 which is widely studied in hemopoiesis. This papers should be revised including the role of this receptor in platelets and megakaryocytes.

G protein-coupled estrogen receptors (GPERs) are expressed throughout the cardiovascular system in humans and animals of both sexes (Meyer et al, Vascul Pharmacol, 2011). To our knowledge, the expression of GPERs has not been investigated in platelets. According to Di vito et al, GPERs are not expressed in mature megacaryocytes.

Following the recommendation of the reviewer, a sentence and a new reference (18) have been added line 57.

18 : Di Vito C, Bergante S, Balduini A, Rastoldo A, Bagarotti A, Surico N, et al. The oestrogen receptor GPER is expressed in human haematopoietic stem cells but not in mature megakaryocytes. Br J Haematol. avr 2010;149(1):150‑2.

Also, the authors mention that "Hematopoietic chimera mice harboring a selective deletion of ERs alpha or beta were used to demonstrate that the effects of a chronic administration of E2 (200μg/kg/day, 3 weeks) were exclusively because of hematopoietic ERalpha". This sentence is confusing since the major estrogen receptor in the hematopoietic system is ER beta. There are papers that refer ER beta as the key player in megakaryocytes (e.g._Leukemia_ volume31, pages945–956 (2017)). The authors should discuss their results at the light of more recent literature including this paper.

We agree with the rewiever that  ERb act as a key player in megakaryopoiesis.

The sentence has now been rewritten (line 162). Our data using ERaKO and ERbKO mice related only to tail bleeding time and thromboembolism.

Minor points: Line 61: "ovarian failure" should be "ovarian aging" or "...due to natural depletion and aging of the finite amount of oocytes".

The manuscript has been edited accordingly (line 22).

Round 2

Reviewer 2 Report

The authors addressed all comments.